# Sound symbolism in Japanese names: Machine learning approaches to gender classification

Chun Hau Ngai[1]*, Alexander J. Kilpatrick[2], Aleksandra Ćwiek[3]

1 East Asian Languages and Cultures Department, Indiana University, Bloomington, Indiana, United States of America, 2 Faculty of International Studies, Nagoya University of Business and Commerce, Nisshin, Aichi, Japan, 3 Leibniz-Centre General Linguistics, Laboratory Phonology, Berlin, Germany

* chngai@iu.edu

**Data Availability Statement:** All Sound Symbolism in Japanese Names: Machine Learning Approaches to Gender Classification files are avaliable from the OSF database: https://osf.io/yrx4u/.

## Abstract

This study investigates the sound symbolic expressions of gender in Japanese names with machine learning algorithms. The main goal of this study is to explore how gender is expressed in the phonemes that make up Japanese names and whether systematic sound-meaning mappings, observed in Indo-European languages, extend to Japanese. In addition to this, this study compares the performance of machine learning algorithms. Random Forest and XGBoost algorithms are trained using the sounds of names and the typical gender of the referents as the dependent variable. Each algorithm is cross-validated using k-fold cross-validation (28 folds) and tested on samples not included in the training cycle. Both algorithms are shown to be reasonably accurate at classifying names into gender categories; however, the XGBoost model performs significantly better than the Random Forest algorithm. Feature importance scores reveal that certain sounds carry gender information. Namely, the voiced bilabial nasal /m/ and voiceless velar consonant /k/ were associated with femininity, and the high front vowel /i/ were associated with masculinity. The association observed for /i/ and /k/ stand contrary to typical patterns found in other languages, suggesting that Japanese is unique in the sound symbolic expression of gender. This study highlights the importance of considering cultural and linguistic nuances in sound symbolism research and underscores the advantage of XGBoost in capturing complex relationships within the data for improved classification accuracy. These findings contribute to the understanding of sound symbolism and gender associations in language.

## Introduction

One of the more astonishing abilities of humans is that we can make use of specific combinations of hums, pops, clicks, and hisses to express imagery with vivid resolution. Arbitrary form-meaning mappings enable the coinage of complex and abstract terms, which in turn remove the limit as to what can be communicated [1]. However, human language also consists of non-arbitrary form-meaning mappings. The non-arbitrary relationship between sounds

**Funding:** The work included in this submission was funded by grants from the Japan Society for the Promotion of Science (Grant Number: 20K13055). This funding sources provided financial support for the fee related to publication. Our third author, Aleksandra Ćwiek was also supported by the German Research Foundation (DFG) (Grant Number: CW 10/1-1).

**Competing interests:** The authors have declared that no competing interests exist.

and meaning, which is often referred to as *sound symbolism*, has been a topic of scientific inquiry since the early 20[th] century. It is important to note that the current paper makes no distinction between sound symbolism and (vocal) iconicity. However, we acknowledge in certain contexts, vocal iconicity is reserved for direct imitation of environmental sounds [2]. One of the most widely studied examples of sound symbolism is the association between certain sounds and the perception of size. For instance, many language speakers associate high front vowels such as [i] and [ɪ] to denote smallness, while low back vowels, such as [a] and [ɔ] are associated with the imagery of largeness [3, 4]. The recent surge in interest in sound symbolism is reflected by the recent number of review articles on the topic [4–7]. In this study, we examine whether non-arbitrary relationships between sound-and-gender exist in Japanese names by making use of machine learning algorithms to classify Japanese first names into binary gender categories. Here, we specifically examine two machine learning algorithms: the Random Forest algorithm and an extreme gradient boosted algorithm (XGBoost). The classification accuracy of each algorithm is tested to determine whether gender is expressed sound symbolically in Japanese names. Following this, feature importance is examined to determine whether associations previously reported in English are found in Japanese since if they are, this would suggest that the systematic sound symbolic expressions of gender are universal.

Sound symbolism refers to the non-arbitrary associations between sounds, or sequences of sounds, and meaning in speech. Although, it should be noted that the term sound symbolism can be misleading given that symbolism, by definition, implies an arbitrary relationship between form and meaning. Early experiments in sound symbolism made use of pseudowords to examine the psychology of iconic mapping. In a seminal study, Sapir [4] found that 96% of English speakers judged /mal/ to be associated with a larger object and /mil/ with a smaller object. More recently, studies have expanded beyond English speaking populations, and found that the same association was found in Thai, Mandarin, Korean, and Japanese speakers [8–10]. Although it remains disputed as to whether sound-to-size mappings corresponds to vowel height or vowel backness, Shinohara and Kawahara [9] found an effect of vowel backness in size ratings in Japanese speakers. Back vowels and voiced obstruents invoked a larger imagery than those of front vowels and voiceless obstruents. This is also supported by typological survey reported that [i] is often used to express diminutive meanings and smallness [11, 12]. Other than the case of the Bahnar language–for which the opposite is true [13],–low vowels are typically associated with large entities and high vowels are associated with small entities. In the present study, we train machine learning algorithms using the sounds of given names to classify gender into binary gender categories. Aligning with gender stereotypes, we predict that sound that typically reflect smallness will be important in the algorithms for female classification while those that typically reflect largeness will contribute to male classification. Additionally, we also predict that sounds typically associated with femineity (e.g., /m/) contribute to female classification.

The systematic association between sounds and size has been proposed to have a biological basis [14, 15]. Species often convey their size by manipulating the fundamental frequencies (F0) of their vocalization, as their advertisement of size could potentially deceive adversaries [16–18]. Various animals, including birds [19], frogs [20], and mammals [21] have been observed to adjust their vocal pitch to convey submission or aggression. In the context of human vocalization, Ohala [14, 15] argues that fundamental frequencies are influenced by body size, thus establishing an association between sounds and the dimensional aspects of size. Specifically, high-frequency sounds are associated with smallness, while low-frequency sounds are associated with largeness. However, in the context of human speech, formant frequencies, not fundamental frequencies, correlate with body size [22–24]. While vowels exhibit variations across multiple dimensions (e.g., third formant and fundamental frequencies), back vowels

generally have lower formant frequencies compared to front vowels [12, 24, 25]. In line with the frequency code hypothesis, vowels with higher second formant, such as [i], are often associated with smaller concepts, while vowels with low second formants, like [a], tend to convey larger concepts[11, 26, 27].This pattern is observed across languages worldwide, where words expressing *smallness* are disproportionately represented by high vowels, such as [i]. Whereas words conveying the idea of *largeness* tend to feature low back vowels such as [a] and [o] [11, 26, 27].

The application of the frequency code to consonants remains a puzzle, given that formant frequencies fail to provide informative cues regarding the manner and place of articulation of consonants. Moreover, disparities emerge in the consonant-to-size mappings observed in fictional creations [9, 28–31] and those examining existing given names [32–35]. While all these studies reference the frequency code as the underlying factor driving observed patterns, patterns reported in these studies vary across these investigations. Notably, Shinohara and Kawahara [9] observed that the invoked imagery depends on the voicing feature of obstruents, where voiced obstruents evoke a larger imagery compared to voiceless ones. This phenomenon has been attributed to the lower fundamental frequencies in vowels adjacent to voiced obstruents [36, 38, 39]. This is evident by the disproportionate number of voice obstruents in heavier, more evolved fictional Pokémon creatures [29, 37]. Although findings from Pokémon names did provide support for the frequency code [14, 15] names of Pokémon are not entirely equivalent to natural language words or names, as they are often created to highlight certain characteristics of the creatures and do not undergo changes. Conversely, studies investigating English first names have indicated a higher prevalence of sonorants in female names, while obstruents (both voiced and voiceless) tend to be more frequent in male names [35, 38, 39]. It has been hypothesized that the sound-to-gender association arises indirectly through size-sound associations, as female names tend to favor consonants associated with smallness, while male names exhibit a preference for consonants associated with largeness. Nevertheless, to date, no studies have endeavored to reconcile the discrepancies observed in these studies. As a result, it remains an open question how the frequency code [14, 15] is applicable to consonants.

Nevertheless, given names in English have been posited to follow the frequency code [33, 40]. Although people do not generally pick their own name, it has been theorized that parents are drawn to names with desirable stereotype for a given gender [41]. Of specific interest is the physical characteristics of body size. It was widely documented that on average, taller statured men have greater success in reproduction [42, 43]. Conversely, shorter and slimmer women are perceived as more fecund and attractive [44–47]. As a result, parents pick phonemes associated with the desirable stereotype of each gender for their offspring. For example, Pitcher and collaborators [33] systematically examined thousands of popular American and Australian English names. In accordance to the frequency code [14, 15], vowels with high resonating frequencies (e.g., /i/ and /e/) were commonly attested in female names, and while low frequency vowels (e.g., /u/ and /o/) were found in male names. Studies have reported systematic sound-meaning mappings in English personal names [32, 33, 35, 38, 40, 41, 48, 49]. These studies have found systematic prosodic-phonological patterns in names of different genders which includes 'consonant sonority', 'quality of stressed vowels', 'number of syllables', 'number of phonemes', and 'stress location' (see Table 1 for details). These studies have reported that females names are more likely to contain more phonemes and syllables, less likely to stress the initial syllable [34, 47, 50]. For example, the name 'Cecila' /sɪsiljəˈ / fits the criteria of a typical female name. It has four syllables, non-initial stressed syllable, and a high front vowel. The opposite is true for the male name 'Tom', /tɑm/. This name has one closed syllable and a low back vowel.

**Table 1. Systematic prosodic-phonological patterns previously reported in English names [34, 35, 47].**

|  | Male names | Female names |
|---|---|---|
| Consonant sonority | More obstruents | More sonorants |
| Quality of stressed vowels | More palatal vowels | More velar vowels |
| Number of phonemes | Less phonemes | More phonemes |
| Number of syllables | Fewer syllables | More syllables |
| Stress location | More initial stress | More non-initial stress |

These patterns are generally true of non-English languages. For example, Suire and collaborators [40] investigated vowel and consonant patterns in the most popular female and male multisyllabic French names. They found that the number of voiced plosives in the first syllable, place of articulation of vowels, nasality, and number of voiceless fricatives were significant predictors of gender. Male names were more likely to contain back vowels (e.g., /o/ and /ɔ/), nasal vowels (e.g., /ã/ and /ɔ̃/), and voiceless fricatives (e.g., /s/ and /ʃ/). Recently, studies have expanded to cross language comparison. Ackermann and Zimmer [32] examined cross-linguistics systematic sound-gender patterns in first names from 13 countries. Critically, this included languages that are geographically and culturally distant as well as typologically unrelated, such as Mandarin, Turkish, Japanese, Romanian, German, Hebrew, among others. They found that the number of non-palatal vowels did not interact with the factor country, suggesting that only patterns pertaining to vowels are sound symbolic. Taken together, these studies illustrated that sound-gender patterns for vowels are robustly attested across languages. In comparison, patterns observed for consonants have limited generalizability.

Our study examines given names in the context of Japanese. Japanese names are multi-syllabic, composed of Kanji, Kana, or a mix of both. Latin alphabets and numerals are not used in given names. For the two writing systems, Kanji characters are logographic characters adopted from early Chinese writing; Kana, on the other hand, is a collective term referred to the two alphabets, Hiragana and Katakana [51]. Although Hiragana and Katakana are relatively phonemic, Kanji characters are not bound to a single pronunciation [52]. Instead, Kanji characters have at least two readings: Kun-reading (the original Japanese reading), and On-reading (reading derived from importation of the Chinese characters into the written Japanese language [53]). For example, the pronunciation of the female name 紀子 is completely ambiguous, it could be pronounced as *Kiko (/kʲikọ/)*, *Toshiko (/toɕikọ/)*, *Michiko (/mʲi kʲikọ/)*, *Motoko (/mọtọkọ/)*. Although gender is not morphologically marked in Japanese, certain suffixes are commonly associated with male or female names. The full list of commonly found elements includes *-o, -ro, -to, -hiko, -ta, -shi* for male names, and *-ko, -e, -yo, -ka, and -mi* for female names [51]. Given the rich inventory of mimetics [54] and iconic mappings in fictional characters in Japanese [54], we expect systematic sound-gender mapping in Japanese names.

Japanese is a member of the Japonic language family, together with Ryukyuan and Hachijō. Although not limited to Japanese, one of the distinct features of Japanese are its use of mora while phonemically contrasting duration. Mora (symbolized μ) is a sub-syllabic timing unit, composed of an optional consonant and a vowel [55]. The assignment of mora is sensitive to the phonological notion of weights. Other than onset consonant, short vowels and coda consonants receive one mora, with long vowels receive two morae. For example, the loanword London /ro.n.do.n/ would consist of four mora. The full consonant and vowel inventories can be found in Table 2 and Fig 1. These inventories make up the features in the datasets except for the velar nasal. This was done to align with literature on Japanese where all instances of coda nasals were counted as /ɴ/.

**Table 2. Consonant inventory of Japanese [56].**

|  | Bilabial | Alveolar | Alveo-palatal | Palatal | Velar | Uvular | Glottal |
|---|---|---|---|---|---|---|---|
| **Plosive** | b p | t d |  |  | k g |  |  |
| **Nasals** | m | n |  |  |  | ɴ |  |
| **Tap** |  | ɾ |  |  |  |  |  |
| **Fricatives** |  | s z | ʃ |  |  |  | h |
| **Affricates** |  | t͡s d͡z | t͡ʃ |  |  |  |  |
| **Glides** | w |  |  | j |  |  |  |

Beyond its moraic structure, Japanese is also known for its rich inventory of mimetics, unlike Indo-European languages. Mimetics, also known as ideophones, are vividly depictions of sensations, actions, and various subjective experiences through iconic sound-meaning correspondences [57]. Since these words are often poorly integrated grammatically, they are often considered a special part of the lexicon [58]. Examples that have been noted include mimetics like goro-goro for sounds of rolling and pika-pika for shiny or flashing object [59]. Given mimetics' integral role in Japanese, it is likely that sound-gender association is encoded in names, another core domain of the lexicon.

Other than unraveling the sound-gender association in Japanese, the current study also aims to test Random Forest algorithms against XGBoost algorithms in classifying names to the typical gender of referents according to phonemes. Although past studies have compared accuracy between these algorithms [60, 61], it remains challenging to ascertain which model performs better because only a single iteration of each model was constructed. To address this issue, this study constructed multiple iterations of each algorithm through k-fold cross-validation that allows for traditional statistical hypothesis testing in the form of linear regression. The Random Forest algorithm [62] and the XGBoost algorithm [63] are both ensemble

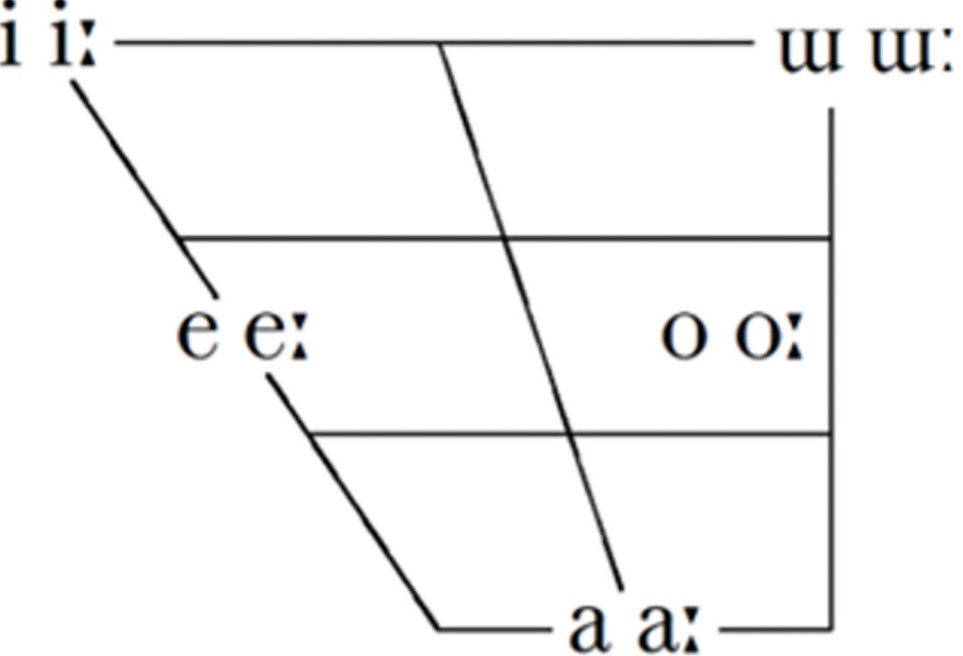

**Fig 1. Vowel inventory of Japanese [56].**

machine learning algorithms that construct many decision trees to generate a model which is tested on a holdout subset of the data. Decision trees are non-parametric machine learning algorithms that resemble flow charts that map the possible outcomes given a series of choices. The choices occur at nodes in the decision tree and the outcome is determined at a terminal node by majority vote in classification models. Random Forest and XGBoost algorithms differ in a few keyways. Although both algorithms construct decision trees, XGBoost models construct sequential trees that take the results of earlier trees into consideration, while Random Forest construct parallel trees independent of each other. In XGBoost, weaker decision trees are trained on the residuals or errors of stronger decision trees. This process involves emphasizing areas where proficient decision trees exhibit deficiencies, with the goal of rectifying those specific errors. This collaborative optimization contributes to the overall model by refining its ability to address diverse scenarios and minimizing prediction errors.

## Method and materials

All data and codes are available at the following online repository: https://osf.io/yrx4u/?view_only=d7fc5ef8b4ab449d8aaba591b18fdfc9.

### Data

The data consists of the 1000 most common given names in Japanese from the Forebears website (REF: https://forebears.io/japan/forenames). Japanese names listed on Forebears are only listed in roman characters and are reportedly converted from their native script using JTALK (REF: https://forebears.io/japan/forenames). The moderators of the website revealed that the names were taken from a 2014 telephone directory consisting of 18 million names. Gender is listed as a distribution between male and female. The data was inspected by a native Japanese linguist. For this study, names were assigned to gender categories through a majority split of their distribution. Four names that were not gender tagged were omitted from the final dataset, resulting in 996 samples (444 female). Since Hiragana and Katakana are reasonably phonemic, an algorithm was constructed to convert Japanese names into phone counts, the output of the algorithm was checked by the same native Japanese linguist. The linguist checked to see if the transcription and gender classification was accurate. They reported that the transcription was accurate and that, overall, gender classifications were accurate, though they did note a couple of cases of unisex names whose majority classification did not meet their expectations. Given that this was a very small number of edge cases, we made no adjustments to the gender distribution listed on the website. Each sample therefore consisted of a gender classification followed by 34 features. The name *Hiromi* /hiɾomi/ for instance is assigned to the female category because of having a 92% distribution to that gender. It consists of value of 2 for /i/, and a value of 1 each for /h/, /ɾ/, /o/, and /m/. A value of zero is applied to all other features. This method results in a dataset primarily consisting of null values (84.32%).

### Machine learning algorithms

The machine learning algorithms outlined in the following were constructed in R. The Random Forest algorithms were constructed with the Ranger package [61, 64] and tuned using the tuneRanger package [65]. The XGBoost algorithms were constructed using the XGBoost package [63] and tuned by inputting different operators to the XGBoost tuning grid. The only hyperparameter that was not tuned in this manner is the number of trees (or rounds in the parlance of XGBoost) for each algorithm. Given that we are comparing between algorithms, other hyperparameters and algorithm features were identical. A series of test models for both

algorithms suggested that the stability and accuracy of the models did not improve after 3000 trees. This value was applied to all iterations.

Because decision tree-based algorithms are prone to overfitting when dealing with datasets that have many null values [56], k-fold cross-validation was used. In k-fold cross validation, the data is split into folds which are then recombined to multiple testing and training subsets. The following algorithms consist of multiple k-fold models that use different combinations of samples which are balanced so that each sample occurs in both the testing and training subsets an even number of times. In the present study, the data was split into 8 folds (A-H). These were then recombined to create subset splits consisting of a 3:1 split to training and testing subsets, whereby each iteration is trained using 75% of the data and tested on the remaining 25%. For example, the first iteration of each model is trained on subsets A, B, C, D, E, and F, and tested on subsets G and H. A Latin square combining all subsets revealed 28 possible combinations, and each combination was used resulting in 28 iterations for each algorithm. 28 folds was selected in order to ensure an adequate sample size for the statistical tests that explore accuracy differences between the Random Forest and XGBoost algorithms. A 3:1 split was determined suitable as the documentation for the Random Forest algorithm suggests a 2:1 split, while the documentation for the XGBoost algorithm suggests a 4:1 split.

Aside from cross-validation, k-folds allow for more in-depth statistical analyses. For example, the present study constructs 28 iterations of both Random Forest and XGBoost algorithms, and we are interested in testing which algorithm is more suited to the task. Single iterations of each model will give a general idea as to how well each algorithm performs specific to the subset splits, but multiple iterations provide mean and standard deviation, and show how the algorithms perform when dealing with the entirety of the data. This also means that statistical hypothesis testing could be applied using the accuracy of the two algorithms, such as the regression analyses presented in the following section. Both algorithms already have in-built significance tests. For Random Forest models, statistical significance tests are applied to feature importance while for XGBoost models, significant tests are applied to the accuracy of the model. We use Fisher's method [66] for combining $p$ values to provide an overall significance test for the model.

## Results

### Accuracy

Overall, the Random Forest algorithm ($M = 76.36\%$, $SD = 1.82\%$) was slightly less accurate and less stable than the XGBoost algorithm ($M = 77.16\%$, $SD = 1.58\%$). The accuracy for all iterations of the XGBoost algorithm were statistically significant ($p < 0.001$ in all cases) as was the Fisher's method combined $p$ value ($p < 0.001$). Because the ranger package does not provide a significance test for Random Forest accuracy, an intercept only linear regression model was constructed using the accuracy of each iteration against 55.42% because that is the accuracy that a naïve model would achieve if it assigned samples to the majority category (55.42% of the samples are male). The intercept only model showed the accuracy of the Random Forest algorithms to be significant; $t(27) = 59.92$, $p < 0.001$. Both models achieved an average classification accuracy of greater than 75%. Table 3 presents the confusion matrix for the Random Forest algorithm and Table 4 presents the confusion matrix for the XGBoost algorithm.

To examine whether the algorithms differ significantly in classifying names, a simple linear regression model was constructed based on the accuracy of individual iterations. It was constructed with models (Random Forest or XGBoost) as the predictor and accuracy as the outcome variable. The simple linear model revealed that the XGBoost algorithm was significantly

**Table 3. Confusion matrix for all iterations of the Random Forest algorithm.**

| . | | Prediction | |
|---|---|---|---|
| | | Female | Male |
| **Sample** | **Female** | 2212 | 896 |
| | **Male** | 752 | 3112 |

**Table 4. Confusion matrix for all iterations of the XGBoost algorithm.**

| | | Classification | |
|---|---|---|---|
| | | Female | Male |
| **Sample** | **Female** | 2303 | 805 |
| | **Male** | 787 | 3077 |

more accurate than the Random Forest algorithm in classifying names according to gender; $t(54) = 2.159$, $p = 0.035$.

## Feature importance

Feature importance is a measure of how important each sound was in the decision-making processes of the machine learning algorithms. For the Random Forest models, the present study uses the Altmann method [67] method of permutation which involves conducing many iterations for each permutation. Permutation is the randomisation of each individual feature and the observation of change in predicting unseen sample due to randomisation. The aggregated feature importance scores for the 10 most important features in the Random Forest model are presented in Table 5. Aggregated permutation importance of each feature and Fisher's combined *p*-values are also listed as *Importance* and *Sig* in Table 5. Table 5 also lists the adjusted distribution to the female category. Because of the uneven distribution of male and female names in the sample (Female = 444; Male = 552), percentage of each phoneme in female distribution is adjusted below. Phoneme with adjusted distribution higher than 50% suggest the particular phoneme predicted female names, and opposite for male names. Since the majority of features that were found to be predictive of gender classification overlapped

**Table 5. Feature importance for the Random Forest model.** The adjusted distribution signifies that if the value is above 50%, it is predicted to be particularly important for female names.

| Phoneme | Importance | Sig | Adjusted Distribution (Female) |
|---|---|---|---|
| /i/ | 5.94% | < 0.001 | 44.97% |
| /m/ | 3.14% | < 0.001 | 78.67% |
| /a/ | 2.16% | < 0.001 | 50.83% |
| /o/ | 1.99% | < 0.001 | 50.29% |
| /k/ | 1.72% | < 0.001 | 61.69% |
| /d͡z/ | 1.51% | < 0.001 | 22.16% |
| /u/ | 1.19% | < 0.001 | 52.13% |
| /ʃ/ | 1.03% | < 0.001 | 30.60% |
| /t/ | 1.00% | < 0.001 | 38.85% |
| /e/ | 0.70% | < 0.001 | 69.48% |
| /h/ | 0.53% | < 0.001 | 33.41% |
| /b/ | 0.45% | < 0.001 | 43.51% |

**Table 6. Feature importance for the XGBoost model.** The adjusted distribution signifies that if the value is above 50%, it is predicted to be particularly important for female names.

| Phoneme | Importance | Adjusted Distribution (Female) |
|---------|------------|-------------------------------|
| /i/ | 99.78 | 44.97% |
| /a/ | 73.25 | 50.83% |
| /m/ | 52.39 | 78.67% |
| /o/ | 46.33 | 50.29% |
| /k/ | 46.17 | 61.69% |
| /u/ | 42.43 | 52.13% |
| /d͡z/ | 33.65 | 22.16% |
| /ʃ/ | 26.20 | 30.60% |
| /s/ | 26.20 | 51.04% |
| /e/ | 25.25 | 69.48% |
| /t/ | 24.04 | 38.85% |
| /h/ | 21.03 | 33.41% |

between Random Forest and XGBoost, features predictive of gender are discussed collectively in the following section.

The XGBoost, feature importance was calculated using the default settings included in the XGBoost package. Feature importance measures are reported in relation to the most important feature in each model. In other words, the most important feature is assigned a score of 100 and each other feature is assigned a score relative to it. Importance is calculated using purity (Gini index) applied to the amount that each attribute split improves model performance, weighted by the number of observations at the node. Table 6 reports the ten most important features in the XGBoost model. *Importance* and *Sig* in Table 6 are the aggregated permutation importance of each feature and the combined *p* value respectively.

Features that were found to be important in the Random Forest model were also found to be important in the XGBoost model. Although the degree to which phonemes contribute to gender classification differ between algorithms, the same set of vowels and consonants could still be found; five phonemes were especially important: /i/, /m/, /a/, /o/ and /k/.

The directionality of phoneme was also examined, and did not entirely align with predictions put forth by the frequency code. Based on previous studies on English names, one would predict that sonorants and high vowels are more commonly found in female names, while obstruents and low vowels are more likely to be found in male names [37, 41]. Within the five most important phonemes, only the voiced bilabial nasal /m/ (a sonorant) aligns with associations reported in English, and was found to be predictive of female names. Opposite to associations observed in English, the high front vowel /i/ and voiceless velar stop /k/ were found to be predictive of male and female names respectively. Potential explanations behind these anomalies are further explored in a post-hoc poisson regression. For phonemes /a/ and /o/, since there is a roughly equal distribution of these phonemes across male and female names, the current study abstains from drawing any conclusion about the directionality of these phonemes.

## Post-hoc analysis

Since associations observed for /i/ and /k/ deviate from patterns previously reported in English, a post-hoc poisson regression analysis was conducted to explore tentative explanations. Here, we proposed that the association observed was masked by suffixations in names. In the case of Japanese, certain phonemic combinations are systematically found in names of a particular gender [53]. In Japanese, the triphone /itʃi/ and the diphone /ko/ could often be

found in male and female names correspondingly. The triphone /itʃi/ corresponds to the kanji 一, which translate to the number one [52]. Whereas for the diphone /ko/, it corresponds to the child character 子 (or 'young') and is frequently found in word-final position of female names [68]. A post-hoc logistical Poisson regression confirms our hypothesis. The triphone /itʃi/ was found to significantly predict male names ($B$ = -2.13, $SE$ = 0.423, $p < 0.001$). The diphone /ko/ was found to significantly predict female names ($B$ = 1.32, $SE$ = 0.202, $p < 0.001$). Both observations were supplemented by many names consisted of itʃi/ (e.g., Ichini /itʃini/, Koichi /koitʃi/, and Shinichi /ʃinitʃi/) in the current dataset and /ko/ (e.g., Akiko /akiko/, Yukiko /yukiko/, and Yoko /yoko/). Given these results, it was therefore likely that associations for /k/ and /i/ were in fact masked by a phonestheme. Here phonestheme is defined as systematic sound-meaning mappings due to shared genealogy [68]. A commonly cited example is the phoneme sequence gl- in English words related to light or vision (e.g., glitter, gleam, and glare, etc.) [5, 69].

In addition to examining phonesthemic explanation, the potential effect of position was also explored here, since Ackermann and Zimmer [34] reported a bifuraction in the sound-gender association for non-palatal vowels (e.g., /a/) in word-final position, as opposed to word initial or word-medial position. Contrary to Ackermann and Zimmer [34], a position effect was not attested. Upon excluding the combination of /itʃi/, predictors /i/ in word-final position ($B$ = -0.293, $SE$ = 0.174, $z$ = -1.684, $p$ = 0.092) and non-word final position ($B$ = -0.288, $SE$ = 0.167, $z$ = -1.172, $p$ = 0.085) did not predict names of either gender.

## Discussion

Due to the sparsity of studies examining sound symbolism in the context of names, most studies have been limited to examining names from Indo-European languages [35, 47, 48]. Nevertheless, our current results demonstrate that sound symbolism sound-gender associations reported in Indo-European languages are also prevalent in Japanese names. With only phonological information, both machine learning algorithms were able to capitalize on the systematic sound-meaning mapping in classifying the gender of names with well above average accuracy (Random Forest: $M$ = 76.36%, $SD$ = 1.82%; XGBoost: $M$ = 77.16%, $SD$ = 1.58%). The high accuracy in gender prediction suggests that sound-symbolic mapping could also be attained in Japanese and could be detected by machine learning algorithms.

In addition to the observed sound-gender associations in Japanese names, our findings may be further understood in the context of the bouba-kiki effect. This phenomenon, where people non-arbitrarily associate certain sounds with specific shapes [70], may extend to the perception of names. The systematic sound-meaning mappings we observed could reflect a broader linguistic phenomenon where certain phonetic qualities are generally associated with specific semantic properties, as discussed in Sidhu, Westbury, Hollis, and Pexman [71].

Of particular interest is the mapping between specific phonemes to male or female names. The sound-gender association observed for voiced bilabial nasal /m/ aligned with those previously reported in English. In previous accounts on English and French, sonorants and front vowels were more frequently attested in female names while obstruents and back vowels were more frequently found in male names [33, 40, 48], an observation in line with Ohala's [14, 15] frequency code. Given that /m/ is a sonorant, its associationwith femininity was not suprising.

The observed association of the consonant /m/ with femininity could be attributed not only to the frequency code [14, 15], but potentially to the idea of the concept of *breast* [11] or *cuteness* [72, 73]. This could be attributed to the articulatory gesture involved. To articulate the consonant /m/, it requires the closure of both lips, and air flowing through the nasal cavity (i.e. bilabial and nasality). Since /m/ was the type of sound that was generated during breastfeeding,

an association was formed between breast and /m/ [26, 74] Alternatively, /m/ have been found to be associated with cuteness in Japanese [72, 73] The consonant /m/ is widely observed in brand names related to baby diapers [72]. This finding also resonates with a study by Erben Johansson & Cronhamn [75] who found that nasals occur more frequently in feminine nouns. Considering the rich imagery invoked by the consonant /m/, the association between the consonant /m/ to femininity may be due to a combination of these imagery or concepts invoked. The bilabial and nasal qualities of /m/, reminiscent of nurturing and cuteness, may cross-linguistically invoke certain imagery or concepts, transcending cultural and linguistic boundaries.

Beyond the consonant /m/, the current study also observed associations that run contrary to those reported in English. Namely, the high front vowel /i/ and the voiceless velar consonant /k/ were found to be associated with masculinity and femininity, respectively. Morphology is likely to play a crucial role in explaining these findings, as Japanese does not overtly mark gender grammatically; however, certain name endings subtly provide indications of the bearer's gender. It was found that the Japanese speakers rely on name-final syllables such as [ta], [to], [ma], [shi], [o], and [ro] to predict male names, and [ko], [ne], [ka], [e], [mi], [yo], [no], and [na] to predicts female name [76]. These elements could sometimes be classified as morphemes since they adhere to Japanese mora structure.

While investigating the associations between gender and phonemes in Japanese names, the current study is hesitant to draw conclusions regarding the open front vowel /a/ and the close-mid back vowel /o/. The reluctance stems from the near 50 percent distribution of these vowels in both male and female names in the training subset, making it unlikely that any observation may not be substantial and could fail to replicate. While phonemes have an equal distribution across genders, it becomes challenging to discern clear sound-symbolic associations. In such cases, small variation in the data or the inclusion of additional names could potentially lead to different results. Therefore, caution is warranted in interpreting the significance of these vowels in predicting genders in names, and a larger sample size may be needed to confirm or refute potential associations.

As for comparing the performance of the two algorithms, the small but significantly higher classification accuracy observed for the XGBoost algorithm provides valuable implication for future studies comparing performance between machine learning algorithms. The superior performance of the XGBoost algorithm suggests that the boosting technique, which optimizes an objective function through sequential tree-building, better represents the non-linear relationships within the data. The higher accuracy achieved by the XGBoost algorithm indicates its ability to discern intricate patterns and interactions among features, which might have been challenging for the Random Forest algorithm to capture. By outperforming the Random Forest algorithm, the XGBoost algorithm demonstrates its potential to yield more precise and reliable predictions, making it a promising choice for similar classification tasks in the future. Nevertheless, careful scrutiny of specific features and relationships is needed to gain insight behind the emergence of improved accuracy.

In terms of the broader discourse on ethical considerations in machine learning, we acknowledge the imperative to explicitly address the ethical implications of our gender classification model based on Japanese name phonemes. While our primary focus has been on exploring how sounds express gender in Japanese, we recognize the importance of engaging with the ethical dimensions surrounding AI applications, especially those related to gender. Concerns regarding potential biases and misuse underscore the need for ongoing vigilance in the development and deployment of such models. In addition to these concerns, it is crucial to highlight that our model operates within a binary classification system of male and female, reflecting the specific context of Japanese names and their phonetic attributes. We

acknowledge the limitation of our model's accuracy in capturing the evolving nuances of gender identity, as societal perspectives continue to shift towards a more nuanced understanding. In subsequent research, we aim to explore approaches that accommodate the complexities of gender identity and contribute to a more inclusive and ethically responsible use of machine learning in this domain. In conclusion, our study provides compelling evidence that sound symbolic sound-gender associations extend to Japanese names. The classification accuracy achieved by both machine learning algorithms, particularly the XGBoost algorithm, highlights the potential for systematic sound-meaning mappings in Japanese, detectable by these algorithms. Notably, we observed intriguing associations between specific phonemes and gender in Japanese names. The sound-gender correspondence for the voiced bilabial nasal /m/ aligns with findings from English and French, possibly linked to the frequency code and associations with *breast* or *cuteness*. However, the associations for the high front vowel /i/ and the voiceless velar consonant /k/ diverge from the patterns reported in English, indicating potential cross-linguistic differences perhaps due to morphology. Culture may also play a role, influencing the phoneme choice in names and reflecting shifts in gender expectations over time. However, our study approaches with caution when interpreting the associations between gender and the vowels /a/ and /o/, due to their near-equal distribution in both male and female names, making any conclusions uncertain without a larger sample size. The superior performance of the XGBoost algorithm over the Random Forest algorithm in gender prediction signifies the advantage of boosting techniques in capturing complex relationships within the data, improving accuracy by leveraging previous tree outputs to correct errors. This observation underscores the potential of the XGBoost algorithm for future classification tasks. Further investigation is needed to unveil the specific features and relationships driving the improved accuracy. Our study contributes to the growing understanding of sound symbolism and gender associations in language, while emphasizing the importance of considering cultural and linguistic nuances in such investigations.

## Acknowledgments

We would like to express our sincere gratitude to all those who have contributed to this research. We extend our appreciation to the Forebears website for providing the dataset of Japanese names, as well as the moderators for their assistance in obtaining the data and linguistic insights. We are also thankful to the native Japanese linguist who meticulously inspected the dataset. Additionally, we acknowledge the developers of the R packages (Ranger, tuneRanger, and XGBoost) used in constructing and tuning the machine learning algorithms. Without their valuable contributions, this study would not have been possible.

## Author Contributions

**Conceptualization:** Chun Hau Ngai.

**Data curation:** Alexander J. Kilpatrick.

**Formal analysis:** Chun Hau Ngai, Alexander J. Kilpatrick.

**Funding acquisition:** Alexander J. Kilpatrick.

**Investigation:** Chun Hau Ngai, Alexander J. Kilpatrick.

**Methodology:** Alexander J. Kilpatrick.

**Project administration:** Chun Hau Ngai.

**Resources:** Alexander J. Kilpatrick.

**Software:** Alexander J. Kilpatrick.

**Supervision:** Chun Hau Ngai, Aleksandra Ćwiek.

**Validation:** Alexander J. Kilpatrick.

**Visualization:** Chun Hau Ngai, Alexander J. Kilpatrick.

**Writing – original draft:** Chun Hau Ngai.

**Writing – review & editing:** Chun Hau Ngai, Alexander J. Kilpatrick, Aleksandra Ćwiek.

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
