## [Editor Report · Decision Letter 0]

13 Sep 2023

PONE-D-23-25119Sound Symbolism in Japanese Names: Machine Learning Approaches to Gender ClassificationPLOS ONE

Dear Dr. Ngai,

Thank you for submitting your manuscript to PLOS ONE. After careful consideration, we feel that it has merit but does not fully meet PLOS ONE’s publication criteria as it currently stands. Therefore, we invite you to submit a revised version of the manuscript that addresses the points raised during the review process.

See my comments at the end of this message. Please submit your revised manuscript by Oct 28 2023 11:59PM. If you will need more time than this to complete your revisions, please reply to this message or contact the journal office at plosone@plos.org. Please include the following items when submitting your revised manuscript:A rebuttal letter that responds to each point raised by the academic editor and reviewer(s). You should upload this letter as a separate file labeled 'Response to Reviewers'.A marked-up copy of your manuscript that highlights changes made to the original version. You should upload this as a separate file labeled 'Revised Manuscript with Track Changes'.An unmarked version of your revised paper without tracked changes. You should upload this as a separate file labeled 'Manuscript'.

We look forward to receiving your revised manuscript.

Kind regards,

Søren Wichmann, PhD

Academic Editor

PLOS ONE

Journal Requirements:

5. Please include a new copy of Table 2,4,5 in your manuscript; the current table is difficult to read. Please follow the link for more information: https://blogs.plos.org/plos/2019/06/looking-good-tips-for-creating-your-plos-figures-graphics/

6. Please include a copy of Table 7 and 8 which you refer to in your text on page 15.

7. We note you have included a table to which you do not refer in the text of your manuscript. Please ensure that you refer to Table 9 in your text; if accepted, production will need this reference to link the reader to the Table.

Additional Editor Comments:

I have not sent this out for reviewing yet because you need to provide a cleaner manuscript before bothering reviewers. On the first few pages alone I came across numerous typos and stylistic problems. I got tired of noting these when arriving at p. 8. What I noted up to then is indicated below. I also noted numerous problems in the list of references. See also below. But these are just things I happened to quickly note. It is not the case that you can just take care of these things, you need to be more thorough than that. So you need to carefully revise the manuscript taking care of these presentational issues first. Possibly you need to involve a professional copy-editor. When I get a better presented manuscript I will send it out for review.

Three typos in the abstract:

are shown to reasonably -> are shown to be reasonably

was associated -> were associated

and which -> ???

p. 4, clumsy formulation: Combined, evidence suggests a strong tendency to map certain phonemes with the imagery of size, other than speakers of the Bahnar language (13)

p. 4 could potential deceive -> could potentially deceive

p. 4 correlates with body size -> correlate with body size

p. 6 it remains an open question as to how -> it remains an open question how

p. 6 For example, the name ‘Catherine’: the way this is transcribed (which is wrong) it has four syllables; also, it is stressed on the first syllable, while the text says "non-initial stressed syllable"

p. 7 Kanji characters are logographic characters adopted from early Chinese religious texts: a bit weird statement; they are not somehow extracted from specific texts, but adopted from early Chinese writing

p. 8 Kanji characters is -> Kanji characters are

Ref. 2: what is K.A.?

Ref. 5 is garbled

Ref 11 incomplete

Ref 13 incomplete

Ref 15 incomplete

Ref 23 inconsistent use of capitalization

Ref 38 capitalization

Ref 56 capitalization

Ref 58 Forest -> forests

Ref 60 incomplete?

Ref 63 incomplete

Ref 73 capitalization

Ref 75 extra space

---

## [Author Response · Author response to Decision Letter 0]

24 Oct 2023

Ngai Chun Hau

East Asian Languages and Cultures Department

Indiana University, Bloomington

12th October, 2023

Dr. Søren Wichmann

Academic Editor

PLOS ONE

Dear Dr. Wichmann,

We would like to express our sincere appreciation for the opportunity to revise our manuscript titled "Sound Symbolism in Japanese Names: Machine Learning Approaches to Gender Classification" in response to the valuable feedback provided by the editor. We have carefully considered your comments and suggestions, and we believe that the revised manuscript now adequately addresses the concerns raised prior the review process.

We have made substantial revisions to the manuscript to improve to adherence to PLOS ONE's formatting criteria. Specifically, we have addressed the following key points:

• Presentation and Stylistic Issues: We have thoroughly revised the manuscript to rectify the typos, grammatical errors, stylistic problems, and references pointed out by the editor. We have carefully proofread the entire manuscript, starting from the abstract to the references section, to ensure its accuracy and readability.

• Format Requirements: We have carefully followed PLOS ONE's style requirements, including those for file naming. We have reviewed the PLOS ONE style templates provided at the URLs https://journals.plos.org/plosone/s/file?id=wjVg/PLOSOne_formatting_sample_main_body.pdf and https://journals.plos.org/plosone/s/file?id=ba62/PLOSOne_formatting_sample_title_authors_affiliations.pdf, and we have adjusted the format of our manuscript accordingly.

• Code Sharing: As per PLOS ONE's guidelines on code sharing, we could assure you that we will make the osf repository publicly available once the manuscript is accepted. Currently the code has been uploaded to the public repository https://osf.io/yrx4u/?view_only=d7fc5ef8b4ab449d8aaba591b18fdfc9

• Grant Information and Financial Disclosure: We apologize for the previous discrepancies in the 'Funding Information' and 'Financial Disclosure' sections. We have revised the manuscript to provide accurate grant numbers for the awards we received for our study in the 'Funding Information' section. Furthermore, we have included an updated financial disclosure statement in our cover letter to reflect the necessary changes.

• Tables: We have included new copies of Tables 2, 4, 5, and 7 in the manuscript, following the formatting guidelines provided by PLOS ONE. Additionally, we discovered that table 8 and 9 were a mistake due an error in table and figure naming. There was no table 8 and 9 to begin with. In fact, these tables are table 6 and 7. Table 6 and 7 presented the results obtained from Random Forest and XGBoost which could be found in page 14 and 15 respectively.

• Furthermore, we have attached a photo of Figure 3 in response to the request for additional figure files. This addition provides further clarity and support to our findings.

We would like to emphasize that we have taken great care to address all the comments and suggestions provided by the editor. We believe that the revised manuscript now meets PLOS ONE's publication criteria and contributes significantly to the field of sound symbolism in Japanese names.

Once again, we sincerely appreciate the valuable feedback provided by the editor and the editor. We are confident that the revised manuscript will be well-received by the scientific community. Thank you for your time and consideration.

Sincerely,

Ngai Chun Hau

Indiana University, Bloomington

chngai@iu.edu

---

## [Decision Letter · Decision Letter 1]

20 Nov 2023

PONE-D-23-25119R1Sound Symbolism in Japanese Names: Machine Learning Approaches to Gender ClassificationPLOS ONE

Dear Dr. Ngai,

Thank you for submitting your manuscript to PLOS ONE. After careful consideration, we feel that it has merit but does not fully meet PLOS ONE’s publication criteria as it currently stands. Therefore, we invite you to submit a revised version of the manuscript that addresses the points raised during the review process. The reviewers provide many constructive comments, which are largely complementary, since one reviewer looks more at methods and the other is more focussed on the context of the general study of sound symbolism. I strongly advice paying close attention to the comments of both reviewers. Additionally, below my signature I offer some observations on stylistic issue and typos. Please take those into account as well.

We look forward to receiving your revised manuscript.

Kind regards,

Søren Wichmann, PhD

Academic Editor

PLOS ONE

**Additional Editor Comments:**

References are to line numbers

54-56:

feature importance is examined to determine whether associations previously reported in

English are found in Japanese which suggest that the systematic sound symbolic expressions of

gender are universal.

->

feature importance is examined to determine whether associations previously reported in

English are found in Japanese since if they are, this would suggest that the systematic sound symbolic expressions of

gender are universal.

167: the loanword London /ro.n.do.n/ would be consisted of four mora ->  the loanword London /ro.n.do.n/ would consist of four mora

278: the Altmann methods [65] method -> the Altmann method [65]

325: poison regression -> Poisson regression

332: Poison regression -> Poisson regression

347: typologically unique from -> typologically distinct from

360: femineity -> femininity

369-370: in contrary to -> contrary to

367-378: "Considering that these elements adhere to the mora structure of Japanese, and seldomly encountered in other languages, it is fitting to classify these elements as morphemes." Seems to me a logical non sequitur.

404-406

"The superior performance of the XGBoost algorithm suggests that the boosting technique, which optimizes an

objective function through sequential tree-building, allowing for a more effective capture of

complex non-linear relationships within the data." Maybe "allows for" rather than "allowing for" is meant? Something is wrong with this sentence.

414-415: sound symbolism sound-gender associations extend to Japanese names, a language distinct from Indo-European languages. -> sound symbolic sound-gender associations extend to Japanese names, a language distinct from Indo-European languages.

462-464:

Blasi DE, Hammarström H, Stadler PF, Christiansen MH. Sound-meaning association

biases evidenced across thousands of languages. Proc Natl Acad Sci.

2016;113(39):10818–23.

->

Blasi DE, Wichmann S, Stadler PF, Hammarström H, Christiansen MH. Sound-meaning association

biases evidenced across thousands of languages. Proc Natl Acad Sci.

2016;113(39):10818–23

Reviewers' comments:

Reviewer's Responses to Questions

**Comments to the Author**

1. If the authors have adequately addressed your comments raised in a previous round of review and you feel that this manuscript is now acceptable for publication, you may indicate that here to bypass the “Comments to the Author” section, enter your conflict of interest statement in the “Confidential to Editor” section, and submit your "Accept" recommendation.

Reviewer #1: (No Response)

Reviewer #2: (No Response)

2. Is the manuscript technically sound, and do the data support the conclusions?

Reviewer #1: Partly

Reviewer #2: Yes

3. Has the statistical analysis been performed appropriately and rigorously? 

Reviewer #1: Yes

Reviewer #2: Yes

4. Have the authors made all data underlying the findings in their manuscript fully available?

Reviewer #1: Yes

Reviewer #2: Yes

5. Is the manuscript presented in an intelligible fashion and written in standard English?

Reviewer #1: Yes

Reviewer #2: Yes

6. Review Comments to the Author

Reviewer #1: The motivation of the study needs to be explained better. In some research communities (e.g., NLP, machine learning), there was a huge discussion on the ethics of doing machine learning based gender classification in the current age (e.g., see this online discussion: https://www.reddit.com/r/MachineLearning/comments/qmm6uh/d_ethical_concerns_for_ml_to_predict_race_gender/) I am surprised the paper does not even mention such issues. Further, I found the paper to describe a very baseline approach, taking an existing dataset (whose creation process is unclear), and a vague algorithm for feature extraction (my impression is that this is the novel contribution for this paper because data and models are coming from somewhere else). Hence, I felt it needs major revisions.

Three main issues I found problematic are:

1. The dataset: it is clearly taken as is from the website. ""Gender is listed as a distribution between male and female." - in what data is that distribution calculated and how? You write that this data was inspected by a Japanese linguist. But, what was the inspection? What did you find out? Is the data 100% correct or are there errors? Is it totally obvious to distinguish male and female names in Japanese? In many languages, there are some gender neutral names and it is hard to guess the gender from name alone. What was your strategy for such cases?

2. The features: You write: "an algorithm was constructed to convert Japanese names into phone counts, the output of the algorithm was checked by the same native Japanese linguist" but no details are given. This seems to be main original contribution of this paper, considering that the dataset is external, and machine learning approaches are also standard ones. But there is no further detail on this algorithm. What did the manual checking of the output reveal? Were there any disagreements between the algorithm and human evaluation? How much? What is the accuracy of this phone counting program? They should be discussed.

3. Modeling: Why are only Random Forest and XGBoost chosen? Why not other simpler approaches like nearest neighbors or logisitic regression etc? Considering the size of your dataset, perhaps nearest neighbors would have worked too. In a paper like this, it would be good to see a comparison with more algorithms. In terms of feature selection too, there are approaches to estimate more predictive features irrespective of the algorithm used (e.g., looking for correlation of a feature with the predicted class). I think those should also be explored to understand the data better. The fold split also seems rather arbitrary. Why can't you just do a 5-fold or 10-fold stratified CV like most researchers report in their papers? Other than these, some error analysis showing where the algorithm failed and how to improve over this baseline approach would be good too.

Potential limitations of the paper needs to be discussed too, in my opinion.

Reviewer #2: I find this manuscript both interesting and valuable for iconicity research as it demonstrates how relatively new methodologies can be applied to address the challenge of handling increasingly available large datasets. The argumentation is easy to follow, and the language is clear. The statistical analysis also appears sound, although I am not an expert in machine learning, and all data underlying the findings is fully available. My recommendation is that it should be accepted with some minor but crucial revisions.

Main points:

Throughout

Although drawing universal conclusions from the occurrence of iconicity in names used in a single language can be challenging, I would like to stress that thorough language-specific iconicity studies serve the same purpose as descriptions of poorly documented languages. The compiled data and analyses deepen our knowledge of the diversity of iconicity in the linguistic system. This, in turn, helps us gain a clearer picture of the cross-linguistic situation in the field and guides future studies. With this being said, it is also important to clearly point out that the comparison between Japanese and English/Indo-European languages arises from the relatively sparse material on name iconicity research, which is mostly confined to Indo-European languages. Without such clarification, it might sound like English/Indo-European languages are a default when it comes to expected iconic patterns and naming conventions. For example:

“Our current results demonstrate that sound symbolism sound-gender association is also prevalent in Japanese names, a language that is typologically unique from Indo-European languages.”

“In conclusion, our study provides compelling evidence that sound symbolism sound gender associations extend to Japanese names, a language distinct from Indo-European languages.”

Throughout

The predictions made based on the frequency code are fitting and the authors have included an extensive list of relevant literature. The authors also draw some conclusions based on cross-linguistic patterns, such as in: “The observed association of the consonant /m/ with femininity could be attributed not only to the frequency code [15,16], but potentially to the idea of the concept of breast [12] or cuteness [67,68].” I wonder why these are not used as predictions along with the frequency code.

Iconicity in male and female names might appear binary, but the grounds for their associated sounds do not necessarily have to be of the same origin (cf. /m/ > femininity possibly via iconicity in words for ‘breast,’ etc.). Erben Johansson & Cronhamn (2022) tested the presence of iconicity in nominal classification systems, i.e., gender and classifier systems. Based on 344 languages from 212 language families, they found that morphological markings for masculine gender involved front, central, and back vowels, while markings for femininity involved nasal and stop consonants. While not directly name data, the material touches on the same type of male/female distinctions as the manuscript, and the findings align.

Furthermore, markings for human ~grammatical gender (in human/non-human classification systems) were largely associated with the same sound features as feminine. This raises the question of markedness in female and male names as well.

It would also be possible to connect the findings to the language-specific lexicon in general, and possibly to the bouba-kiki effect. References:

Sidhu, D. M., & Pexman, P. M. (2018). Lonely sensational icons: Semantic neighbourhood density, sensory experience and iconicity. Language, Cognition and Neuroscience, 33(1), 25–31. https://doi.org/10.1080/23273798.2017.1358379

Erben Johansson, N. E., & Cronhamn, S. (2022). Vocal iconicity in nominal classification. Language and Cognition, 15(2), 266-291. https://doi.org/10.1017/langcog.2022.36

Sidhu, D. M., Westbury, C., Hollis, G., & Pexman, P. M. (2021). Sound symbolism shapes the English language: The maluma/takete effect in English nouns. Psychonomic Bulletin & Review, 28(4), 1390–1398. https://doi.org/10.3758/s13423-021-01883-3

Page 8

Since this manuscript focuses on iconicity in Japanese, it is important to define what mimetics/ideophones are and their role in the Japanese language, especially in contrast to their status in other languages. This is crucial, considering the frequent comparisons with English and Indo-European languages throughout the text. Additionally, on page 17, “phonestheme” and “phonesthemic” appear without definition and should be clarified.

Page 10

The data used seems sound and the procedure for evaluation the data is well-described, but the authors do not explain why JTALK was chosen in the first place. For convenience, because JTALK is the only available database of Japanese names, or something else?

Page 19-20

The discussion brings up many relevant factors, including morphology and culture. However, while interesting, I find the connection between naming conventions and improvements in women’s rights too speculative without any data to back it up. To draw any conclusions, a comparison to older Japanese names would be needed. Additionally, a shorter paragraph at the beginning of the manuscript summarizing naming conventions from across the world would help contextualize the analyzed material, especially if the authors wish to maintain the point about potential phonological changes in Japanese names over the last decades. For instance, politically motivated names, as seen in Mandarin Chinese, could provide a useful comparison.

Minor points:

Page 3

I think it would benefit the reader if there were a clear linguistic example of sound symbolism in the first paragraph. Not necessarily an example like “bouba-kiki”, but just an association that is cross-linguistically common and can also be found in English or some other global language for familiarity.

Page 3

Since “iconicity” is used in the manuscript, it should be stated whether the authors consider sound symbolism as the same as (vocal) iconicity to avoid confusion. A distinction does not have to be drawn, but in that case, state that you use the terms interchangeably. I also want to mention that I am glad the authors highlighted that the “symbolism” in “sound symbolism” can be misleading.

Page 4

“thatthe” > “that the”

Page 5 and other places

While iconicity research in Pokémon names across languages has increased significantly in recent years, I think these findings should be contextualized. They are created in a manner that is arguably more deliberate than names and words. Therefore, their informative value about iconicity in language ought to be lesser and/or qualitatively different.

Page 8

“Typologically, Japanese differs from many Indo-European languages in many aspects. Japanese is a member of the Japonic language family. Although not limited to Japanese, one of the distinct features of Japanese are…”

The beginning of this paragraph sounds stunted. If the sentence “Japanese is a member of the Japonic language family” were expanded upon, it would likely flow better into the next sentence. For example, “Japanese is a member of the Japonic language family, together with Ryukyuan and Hachijō”, or something similar.

Page 8-9

“alphabets” > “syllabaries”

Hiragana and Katakana are not alphabets.

Page 9-10

Several technical terms are introduced here without description. While some are explained in the method section, others, such as “weak/strong learners”, are not. It would be helpful for the reader if the authors could add a sentence indicating that these terms will be described in detail later in the manuscript.

Page 12

“A partial Latin square revealed 28 possible combinations of subsets, and each combination was 237 used resulting in 28 iterations for each algorithm.”

This sentence should be explained in more detail for less statistics-savvy readers. Why is a partial Latin square used?

Discussion/conclusion

The post hoc analysis highlights the role of sound combinations, which I believe is crucial for understanding iconicity in spoken language data. I am not suggesting that the authors redo the entire analysis to include both phonemes and all possible phoneme combinations, but adding a sentence or two about the implications for future studies would be beneficial.

7. PLOS authors have the option to publish the peer review history of their article (what does this mean?). If published, this will include your full peer review and any attached files.

Reviewer #1: No

Reviewer #2: **Yes: **Niklas Erben Johansson

---

## [Author Response · Author response to Decision Letter 1]

19 Dec 2023

Ngai Chun Hau

East Asian Languages and Cultures Department

Indiana University, Bloomington

15th December 2023

Dear Editor and Reviewers,

We extend our heartfelt gratitude for the invaluable feedback and insightful observations provided during the review process. Your expertise and guidance have been instrumental in refining our manuscript to its current improved state. Below, we have provided detailed responses to your constructive comments, which we have presented indented for clarity and reference.

1. there was a huge discussion on the ethics of doing machine learning based gender classification in the current age (e.g., see this online discussion: https://www.reddit.com/r/MachineLearning/comments/qmm6uh/d_ethical_concerns_for_ml_to_predict_race_gender/) I am surprised the paper does not even mention such issues. Further, I found the paper to describe a very baseline approach, taking an existing dataset (whose creation process is unclear), and a vague algorithm for feature extraction (my impression is that this is the novel contribution for this paper.

We acknowledge the importance of engaging with the ethical dimensions surrounding AI applications, especially those related to gender. A paragraph has been added in the discussion section. In terms of the broader discourse on ethical considerations in machine learning, we recognize the imperative to explicitly address the ethical implications of our gender classification model based on Japanese name phonemes. While our primary focus has been on exploring how sounds express gender in Japanese, concerns regarding potential biases and misuse underscore the need for ongoing vigilance in the development and deployment of such models. We acknowledge the limitation of our model’s accuracy in capturing the evolving nuances of gender identity, as societal perspectives continue to shift towards a more nuanced understanding. In addition to these concerns, it is crucial to highlight that our model operates within a binary classification system of male and female, reflecting the specific context of Japanese names and their phonetic attributes.

2. The dataset: it is clearly taken as is from the website. "Gender is listed as a distribution between male and female." - in what data is that distribution calculated and how? You write that this data was inspected by a Japanese linguist. But, what was the inspection? What did you find out? Is the data 100% correct or are there errors? Is it totally obvious to distinguish male and female names in Japanese? In many languages, there are some gender-neutral names and it is hard to guess the gender from name alone. What was your strategy for such cases?

Thank you for pointing that out. The gender distribution was calculated based on the data available on the website. The dataset was inspected by a Japanese linguist to ensure the transcription and gender classification were accurate. The linguist reported that the transcription was accurate and that, overall, gender classifications were accurate, though they did note a couple of cases of unisex names whose majority classification did not meet their expectations. Given that this was a very small number of edge cases, we made no adjustments to the gender distribution listed on the website. We used the gender of the entry when available, and when it was not available, entries without gender are discarded. We acknowledge that there are some gender-neutral names in Japanese, and we did our best to ensure that the gender classification was accurate. We hope that this explanation addresses the reviewer’s concerns. 

3. Why are only Random Forest and XGBoost chosen? Why not other simpler approaches like nearest neighbors or logisitic regression etc? Considering the size of your dataset, perhaps nearest neighbors would have worked too. In a paper like this, it would be good to see a comparison with more algorithms. In terms of feature selection too, there are approaches to estimate more predictive features irrespective of the algorithm used (e.g., looking for correlation of a feature with the predicted class). I think those should also be explored to understand the data better. The fold split also seems rather arbitrary. Why can't you just do a 5-fold or 10-fold stratified CV like most researchers report in their papers?

We appreciate the reviewer's thoughtful feedback. While we acknowledge the merit in exploring a broader set of algorithms and alternative fold split strategies, we would like to clarify that the focus of our study was on introducing a novel approach to gender classification based on the Random Forest and XGBoost algorithms which are known for their ability to capture intricate patterns and nonlinear relationships, providing a more sophisticated modelling approach. Given the specific goals and scope of our research, we believe that conducting further tests with simpler algorithms may dilute the primary focus of our study. We are, however, open to incorporating additional discussions on the limitations of our chosen approach in the revised version. We appreciate the reviewer's understanding and continued engagement with our work. In respect to the selection of folds; we created a Latin square that would provide 28 unique subset splits because we wanted to conduct a statistical test that was familiar to the reader (regression) and needed enough samples to make it a fair test. Given the closeness in accuracy (and SD) for the two methods, 5- or 10-fold models would likely not achieve significance. No edits were made to the manuscript.

4. Other than these, some error analysis showing where the algorithm failed and how to improve over this baseline approach would be good too.

Sound symbolism is stochastic, not prescriptive. Thus, it is expected that a certain number of errors are found. Misclassified samples are likely not being misclassified because of modelling errors, but rather because those names do not reflect gender sound symbolically. No edits are made to the manuscript.

5. It is also important to clearly point out that the comparison between Japanese and English/Indo-European languages arises from the relatively sparse material on name iconicity research, which is mostly confined to Indo-European languages. Without such clarification, it might sound like English/Indo-European languages are a default when it comes to expected iconic patterns and naming conventions. For example:“Our current results demonstrate that sound symbolism sound-gender association is also prevalent in Japanese names, a language that is typologically unique from Indo-European languages.”

Thank you for pointing that out. These sentences are rephrased to better reflect the sparsity in the existing literature of names.

6. The predictions made based on the frequency code are fitting and the authors have included an extensive list of relevant literature. The authors also draw some conclusions based on cross-linguistic patterns, such as in: “The observed association of the consonant /m/ with femininity could be attributed not only to the frequency code [15,16], but potentially to the idea of the concept of breast [12] or cuteness [67,68].” I wonder why these are not used as predictions along with the frequency code.

Line 85 was added to include this prediction.

7. Johansson & Cronhamn (2022) tested the presence of iconicity in nominal classification systems, i.e., gender and classifier systems. Based on 344 languages from 212 language families, they found that morphological markings for masculine gender involved front, central, and back vowels, while markings for femininity involved nasal and stop consonants. While not directly name data, the material touches on the same type of male/female distinctions as the manuscript, and the findings align.

Lines 431 and 434 are added to reference existing results to the findings in Johansson & Cronhamn (2022). Indeed, the findings of Johansson & Cronhamn (2022) resonate with our study. Erben Johansson & Cronhamn also found that nasals occur more frequently in feminine nouns. Considering the rich imagery invoked by the consonant /m/, the association between the consonant /m/ to femininity may be due to a combination of these imagery or concepts invoked. The bilabial and nasal qualities of /m/, reminiscent of nurturing and cuteness, may cross-linguistically invoke certain imagery or concepts, transcending cultural and linguistic boundaries. 

8. It would also be possible to connect the findings to the language-specific lexicon in general, and possibly to the bouba-kiki effect. References:

Sidhu, D. M., & Pexman, P. M. (2018). Lonely sensational icons: Semantic neighbourhood density, sensory experience and iconicity. Language, Cognition and Neuroscience, 33(1), 25–31. https://doi.org/10.1080/23273798.2017.1358379

Erben Johansson, N. E., & Cronhamn, S. (2022). Vocal iconicity in nominal classification. Language and Cognition, 15(2), 266-291. https://doi.org/10.1017/langcog.2022.36

Sidhu, D. M., Westbury, C., Hollis, G., & Pexman, P. M. (2021). Sound symbolism shapes the English language: The maluma/takete effect in English nouns. Psychonomic Bulletin & Review, 28(4), 1390–1398. https://doi.org/10.3758/s13423-021-01883-3

Lines 402 to 407, 432, and 434 are added to include a discussion related to the bouba-kiki effect. In addition to the observed sound-gender associations in Japanese names, our findings may be further understood in the context of the bouba-kiki effect. This phenomenon, where people non-arbitrarily associate certain sounds with specific shapes, may extend to the perception of names. The systematic sound-meaning mappings we observed could reflect a broader linguistic phenomenon where certain phonetic qualities are generally associated with specific semantic properties, as discussed in Sidhu, Westbury, Hollis, and Pexman.

9. Since this manuscript focuses on iconicity in Japanese, it is important to define what mimetics/ideophones are and their role in the Japanese language, especially in contrast to their status in other languages. This is crucial, considering the frequent comparisons with English and Indo-European languages throughout the text. 

In response to your comment, we have revised the manuscript to include a more comprehensive explanation of mimetics, also known as ideophones, in the Japanese language. We have highlighted their vivid depictions of sensations, actions, and various subjective experiences through iconic sound-meaning correspondences. We have also emphasized their unique status in the lexicon due to their often poor grammatical integration. Examples such as ‘goro-goro’ for sounds of rolling and ‘pika-pika’ for shiny or flashing objects have been included to illustrate this concept. Furthermore, we have discussed the likelihood of sound-gender associations being encoded in names, another core domain of the lexicon, given the integral role of mimetics in Japanese.

10. Additionally, on page 17, “phonestheme” and “phonesthemic” appear without definition and should be clarified.

Thank you for pointing out the need for clarification of the terms “phonestheme” and “phonesthemic”. We understand the importance of defining key terms for the reader’s comprehension. In response to your comment, we have added a definition of “phonestheme” in the manuscript. It is now defined as systematic sound-meaning mappings due to shared genealogy [69]. To illustrate this concept, we have included a commonly cited example of the phoneme sequence ‘gl-’ in English words related to light or vision, such as glitter, gleam, and glare [6,70].

11. The data used seems sound and the procedure for evaluation the data is well-described, but the authors do not explain why JTALK was chosen in the first place. For convenience, because JTALK is the only available database of Japanese names, or something else?

In response to your comment, we have clarified in the manuscript why JTALK was chosen. The primary reason for this choice is the way the names are listed on the website. Unfortunately, their native script is not presented on the website, limiting our options. Despite this limitation, we found that the gender classifications were generally accurate. We did encounter a few cases of unisex names whose majority classification did not meet our expectations. However, given that these were very few edge cases, we decided not to adjust the gender distribution listed on the website.

12. The discussion brings up many relevant factors, including morphology and culture. However, while interesting, I find the connection between naming conventions and improvements in women’s rights too speculative without any data to back it up. To draw any conclusions, a comparison to older Japanese names would be needed. Additionally, a shorter paragraph at the beginning of the manuscript summarizing naming conventions from across the world would help contextualize the analyzed material, especially if the authors wish to maintain the point about potential phonological changes in Japanese names over the last decades. For instance, politically motivated names, as seen in Mandarin Chinese, could provide a useful comparison.

Thank you for your insightful comment. We agree with your observation about the speculative nature of the connection between naming conventions and improvements in women’s rights. In response to your feedback, we have removed the paragraph in question from the discussion.We understand the importance of providing a solid foundation for our arguments and ensuring that our conclusions are backed by data. We appreciate your suggestion about including a summary of naming conventions from across the world to contextualize the analyzed material. We will take this into consideration for future research.

13. I think it would benefit the reader if there were a clear linguistic example of sound symbolism in the first paragraph. Not necessarily an example like “bouba-kiki”, but just an association that is cross-linguistically common and can also be found in English or some other global language for familiarity.

Thank you for your suggestion to include a clear linguistic example of sound symbolism in the first paragraph. We agree that this would benefit the reader and provide a more immediate understanding of the concept.

In response to your comment, we have added an example of sound symbolism in the introduction. We chose an example that is cross-linguistically common and can be found in English, for familiarity. Specifically, we discussed the association between certain sounds and the perception of size. For instance, many language speakers associate high front vowels such as [i] and [ɪ] with smallness, while low back vowels, such as [a] and [ɔ], are associated with largeness [2,3].

14. Since “iconicity” is used in the manuscript, it should be stated whether the authors consider sound symbolism as the same as (vocal) iconicity to avoid confusion. A distinction does not have to be drawn, but in that case, state that you use the terms interchangeably. I also want to mention that I am glad the authors highlighted that the “symbolism” in “sound symbolism” can be misleading.

Thank you for your comment regarding the use of the terms “iconicity” and “sound symbolism” in our manuscript. We understand the potential for confusion and the importance of clarity in our terminology.

In response to your comment, we have added a footnote to clarify our usage of these terms. We have stated that in the context of our paper, we make no distinction between sound symbolism and (vocal) iconicity. However, we acknowledge that in certain contexts, vocal iconicity is reserved for the direct imitation of environmental sounds [6].

15. While iconicity research in Pokémon names across languages has increased significantly in recent years, I think these findings should be contextualized. They are created in a manner that is arguably more deliberate than names and words. Therefore, their informative value about iconicity in language ought to be lesser and/or qualitatively different.

Thank you for your comment regarding the use of Pokémon names in iconicity research. We agree with your observation that these names are created in a more deliberate manner than natural language words or names, and therefore, their informative value about iconicity in language might be lesser and/or qualitatively different.

In response to your comment, we have added a footnote in our manuscript to contextualize the findings from Pokémon names. We have noted that while these names did provide support for the frequency code[15,16], they are not entirely equivalent to natural language words or names. This is because they are often created to highlight certain characteristics of the creatures and do not undergo changes.

16. “Typologically, Japanese differs from many Indo-European languages in many aspects. Japanese is a member of the Japonic language family. Although not limited to Japanese, one of the distinct features of Japanese are…” The beginning of this paragraph sounds stunted. If the sentence “Japanese is a member of the Japonic language family” were expanded upon, it would likely flow better into the next sentence. For example, “Japanese is a member of the Japonic language family, together with Ryukyuan and Hachijō”, or something similar.

Thank you for your feedback on the flow of our manuscript. We agree that the paragraph you pointed out could be improved for better readability and coherence.

In response to your comment, we have rephrased the sentences in question. We now start the paragraph by acknowledging the limited scope of previous studies on sound symbolism in the context of names, which have mostly focused on Indo-European languages [37,38,50]. We then transition into discussing our current results, which demonstrate that sound symbolism and sound-gender associations reported in Indo-European languages are also prevalent in Japanese names.

17. “alphabets” > “syllabaries” Hiragana and Katakana are not alphabets.

Changed.

18. Several technical terms are introduced here without description. While some are explained in the method section, others, such as “weak/strong learners”, are not. It would be helpful for the reader if the authors could add a sentence indicating that these terms will be described in detail later in the manuscript.

Thank you for your comment regarding the introduction of technical terms in our manuscript. We understand the importance of clear definitions for reader comprehension.

In response to your comment, we have added explanations for the terms “weak/strong learners” in the context of XGBoost. We have clarified that weaker decision trees are trained on the residuals or errors of stronger decision trees. This process emphasizes areas where proficient decision trees exhibit deficiencies, with the goal of rectifying those specific errors. This collaborative optimization contributes to the overall model by refining its ability to address diverse scenarios and minimizing prediction errors.

19. “A partial Latin square revealed 28 possible combinations of subsets, and each combination was 237 used resulting in 28 iterations for each algorithm.” This sentence should be explained in more detail for less statistics-savvy readers. Why is a partial Latin square used?

Thank you for your comment regarding the use of a partial Latin square and the need for more detailed explanation. We understand the importance of making our methodology accessible to readers with varying levels of statistical knowledge.

In response to your comment, we have expanded upon our explanation of the use of k-fold cross-validation and the rationale behind the selection of a 3:1 split for training and testing subsets. We clarified that decision tree-based algorithms are prone to overfitting when dealing with datasets that have many null values [56], hence the use of k-fold cross-validation. In this method, the data is split into folds which are then recombined to create multiple testing and training subsets.

We also explained that our study used a Latin square to combine all subsets, revealing 28 possible combinations. Each combination was used, resulting in 28 iterations for each algorithm. This number of folds was selected to ensure an adequate sample size for the statistical tests that explore accuracy differences between the Random Forest and XGBoost algorithms.

We appreciate your thorough assessment of our work and the time and effort you have invested in ensuring the quality and rigor of our research. All other suggested edits in terms of wording and spelling are also incorporated. Your input has significantly contributed to the enhancement of our manuscript. The second reviewer made methodological suggestions regarding collecting additional data from fictional sources. While we feel that their concerns are valid, we have not undertaken any additional analyses as we feel that this would be outside the scope of the present manuscript. We have, however, addressed their concerns in an additional paragraph in the conclusion section where we discuss future directions for this research.

Thank you once again for your dedication to scholarly excellence.

Sincerely,

Ngai Chun Hau

Indiana University, Bloomington

chngai@iu.edu

---

## [Editor Report · Decision Letter 2]

5 Jan 2024

Sound Symbolism in Japanese Names: Machine Learning Approaches to Gender Classification

PONE-D-23-25119R2

Dear Dr. Ngai,

We’re pleased to inform you that your manuscript has been judged scientifically suitable for publication and will be formally accepted for publication once it meets all outstanding technical requirements.

Kind regards,

Søren Wichmann, PhD

Academic Editor

PLOS ONE
---

## [Editor Report · Acceptance letter]

1 Mar 2024

PONE-D-23-25119R2 

PLOS ONE

Dear Dr. Ngai, 

I'm pleased to inform you that your manuscript has been deemed suitable for publication in PLOS ONE. Congratulations! Your manuscript is now being handed over to our production team.

Kind regards, 

on behalf of

Dr. Søren Wichmann 

Academic Editor

PLOS ONE